# Full 2π tunable phase modulation using avoided crossing of resonances

Ju Young Kim[1,3], Juho Park [1,3], Gregory R. Holdman[2], Jacob T. Heiden [1], Shinho Kim [1], Victor W. Brar [2✉] & Min Seok Jang [1✉]

Active metasurfaces have been proposed as one attractive means of achieving high-resolution spatiotemporal control of optical wavefronts, having applications such as LIDAR and dynamic holography. However, achieving full, dynamic phase control has been elusive in metasurfaces. In this paper, we unveil an electrically tunable metasurface design strategy that operates near the avoided crossing of two resonances, one a spectrally narrow, over-coupled resonance and the other with a high resonance frequency tunability. This strategy displays an unprecedented upper limit of 4π range of dynamic phase modulation with no significant variations in optical amplitude, by enhancing the phase tunability through utilizing two coupled resonances. A proof-of-concept metasurface is justified analytically and verified numerically in an experimentally accessible platform using quasi-bound states in the continuum and graphene plasmon resonances, with results showing a 3π phase modulation capacity with a uniform reflection amplitude of ~0.65.

[1] School of Electrical Engineering, Korea Advanced Institute of Science and Technology, Daejeon 34141, Republic of Korea. [2] Department of Physics, University of Wisconsin-Madison, Madison, WI 53706, USA. [3]These authors contributed equally: Ju Young Kim, Juho Park. ✉email: vbrar@wisc.edu; jang.minseok@kaist.ac.kr

Active control of optical phase over an entire $2\pi$ range with individual pixels is an important milestone in photonics, being one critical component needed for full wavefront modulation. Applications that are dependent on local phase control include wavefront shapers[1–3], holography[4,5], polarization control[6], and beam steering/LIDAR[7,8]. In order to experimentally realize $2\pi$ phase control, liquid crystal-on-silicon spatial light modulators[9,10], and optical phased arrays utilizing MEMS[11] and silicon waveguides[12] have been introduced over the years. However, due to their subwavelength nature and superior response speeds, many groups have recently explored the possibility of using tunable metasurfaces as a means of bestowing spatio-temporal control of optical phase[6–8,13–16]. In these devices, the local phase and/or intensity of reflected or transmitted light is locally controlled by adjusting the resonant response of individual meta-atoms, either by varying their dimensions or—usually in the case of dynamic systems—the index of their constituent materials. In static implementations, this strategy has been utilized to create holographic projections, beam steerers, and flat lenses with large deflection angles and achromatic response at frequencies ranging from the UV to THz[1–5,17–20]. For dynamic systems, meanwhile, full phase control requires materials with tunable indices/geometry in the metasurface to tune the localized dielectric or plasmonic resonances of the meta-atoms, and notable works have implemented various methodologies including free carrier tuning[6,8,13,15,16], electromechanical[14], the use of phase-change materials[21,22], thermo-optic effects[23], and liquid crystals[24]. At a fixed operating frequency, this tuning alters the phase delay that the meta-atom imparts on the incoming light and, for many independently controlled meta-atoms, this strategy allows for optical wavefronts to be reshaped.

Currently, dynamic metasurfaces are faced with two central problems. First, tuning the resonances of meta-atoms does not only affect the phase of an optical wavefront, it also leads to sharp changes in the magnitude of the reflected/transmitted light. This leads to a correlation between the phase and amplitude of the light, which is typically highly non-uniform. This problem is worsened by the fact that most materials with tunable indices are also lossy, which leads to strong absorption in the meta-atoms near their resonant frequency. Second, most tunable materials exhibit only a modest change in the optical index, and they cannot sufficiently tune the resonances of the meta-atoms over the full, desired $2\pi$ range. Consequently, complete tunable coverage of the full $0–2\pi$ phase range while maintaining a constant and significant light amplitude has so far remained elusive. Methods that use multiple control parameters (i.e., voltage gates) have demonstrated theoretically[15] and experimentally[16] that a full $2\pi$ phase shift is possible. However, the multiple control parameters make the operation of these devices complicated and the reflected light amplitudes of those implementations were small and/or non-uniform across all phases[16,25].

In this work, we first discuss the key reasons behind the difficulty in using electro-optic controls with metasurfaces to tune the optical phase over the full $2\pi$ range. We describe the 'tradeoff' problem, where increasing the frequency tunability of the metasurface resonance also increases its spectral linewidth, which limits the effective range of phase tuning. We then propose a solution that circumvents this trade-off by utilizing an avoided crossing (also known as anticrossing) between a spectrally narrow, over-coupled resonance and a resonance with a large resonance frequency shift. This introduces significantly more phase tunability at the anticrossing point than would be possible with standard electro-optic metasurfaces. Finally, we theoretically justify and numerically verify a proof-of-concept metasurface in an experimentally accessible platform utilizing quasi-bound states in the continuum (qBIC) and graphene plasmons that actively modulate a phase range of not just $2\pi$ but over $3\pi$ with uniform amplitude.

## Results

**Conventional limits in phase modulation.** To explain the necessity of utilizing avoided crossing between two resonances, we would like to first point out the difficulties in achieving full $2\pi$ phase modulation using a single resonance. To realize dynamic $2\pi$ phase tuning with uniform amplitude, the resonant metasurface must satisfy the following conditions. First, one should double the usual phase range of $0–\pi$ of single resonances to $0–2\pi$, usually done by using a back reflector for repeated metasurface interaction or by overlapping two resonances through geometrical parameters tuning[6,15]. Second, to receive radiated light from the electrons with a delayed phase without most of it being absorbed, the resonant mode of the metasurface must be over-coupled to the incident light with its radiative loss rate $\gamma_r$ exceeding the dissipative loss rate $\gamma_d$. As described by the temporal coupled-mode theory (TCMT) and illustrated in Fig. 1a, over-coupling brings the center of the circle of the complex amplitude distribution of the scattered light closer to the origin. This allows a highly over-coupled system ($\gamma_r \gg \gamma_d$) to maintain uniform amplitude as the phase is varied[26]. Third, to access the full $2\pi$ phase range, the spectral profile must sweep entirely across the operating frequency (Fig. 1b). These three conditions outline the following simultaneous objectives for metasurface design: Compress the spectral profile by reducing the resonance linewidth, enhance the radiative loss rate and/or decrease the dissipative loss rate, and maximally shift the resonance frequency itself.

However, there lies an inherent trade-off problem for simultaneously achieving a large resonance frequency shift and a narrow spectral linewidth, which consequently limits the achievable phase tuning range below $2\pi$ as illustrated in Fig. 1a, b. The resonance frequency shift, $\triangle\omega$, can be approximated by first-order perturbation theory as follows[26]:

$$\triangle\omega = -\frac{\omega_0}{2}\frac{\int dv \,\triangle\epsilon(\mathbf{r})\left|\mathbf{E}(\mathbf{r})\right|^2}{\int dv \,\epsilon(\mathbf{r})\left|\mathbf{E}(\mathbf{r})\right|^2}. \tag{1}$$

Here $\omega_0$, $\mathbf{E}(\mathbf{r})$ are the resonance frequency of the mode and its electric field, respectively, and $\epsilon(\mathbf{r})$, $\triangle\epsilon(\mathbf{r})$ are the permittivity distribution and the change in permittivity induced by the control parameter modulation, respectively. The integration is over the unit cell containing the meta-atom. On the other hand, the spectral width, or the full-width-half-maximum (FWHM) of the resonance is given by $2(\gamma_r + \gamma_d)$. Since the system must be in over-coupled regime, the ratio of $\gamma_r$ to $\gamma_d$, is greater than unity, $\eta = \gamma_r/\gamma_d > 1$. The dissipative loss rate is written as the following equation[26]:

$$\gamma_d = \frac{\omega_0}{2}\frac{\int dv \,\mathrm{Im}[\epsilon]\left|\mathbf{E}(\mathbf{r})\right|^2}{\int dv \,\epsilon(\mathbf{r})\left|\mathbf{E}(\mathbf{r})\right|^2}. \tag{2}$$

Therefore, the normalized effective resonance frequency shift, $\left|\triangle\omega/\mathrm{FWHM}\right|$, which needs to be maximized to achieve a wide phase tuning range, can be simplified as

$$\left|\frac{\triangle\omega}{\mathrm{FWHM}}\right| = \frac{1}{2(1+\eta)}\frac{\int dv \,\triangle\epsilon(\mathbf{r})\left|\mathbf{E}(\mathbf{r})\right|^2}{\int dv \,\mathrm{Im}[\epsilon(\mathbf{r})]\left|\mathbf{E}(\mathbf{r})\right|^2} \leq \frac{1}{2(1+\eta)}\frac{\triangle\epsilon_{\mathrm{tune}}}{\mathrm{Im}\left[\epsilon_{\mathrm{tune}}\right]}, \tag{3}$$

where the last equality holds for the idealistic case of the dissipation only occurring in the tunable material of permittivity $\epsilon_{\mathrm{tune}}$. For a material that tunes via control of free carrier density (a Drude material), $\epsilon_{\mathrm{tune}} \propto n/(\omega + i\tau^{-1})$ where $n$ and $\tau$ are the carrier density and the carrier relaxation time, respectively.

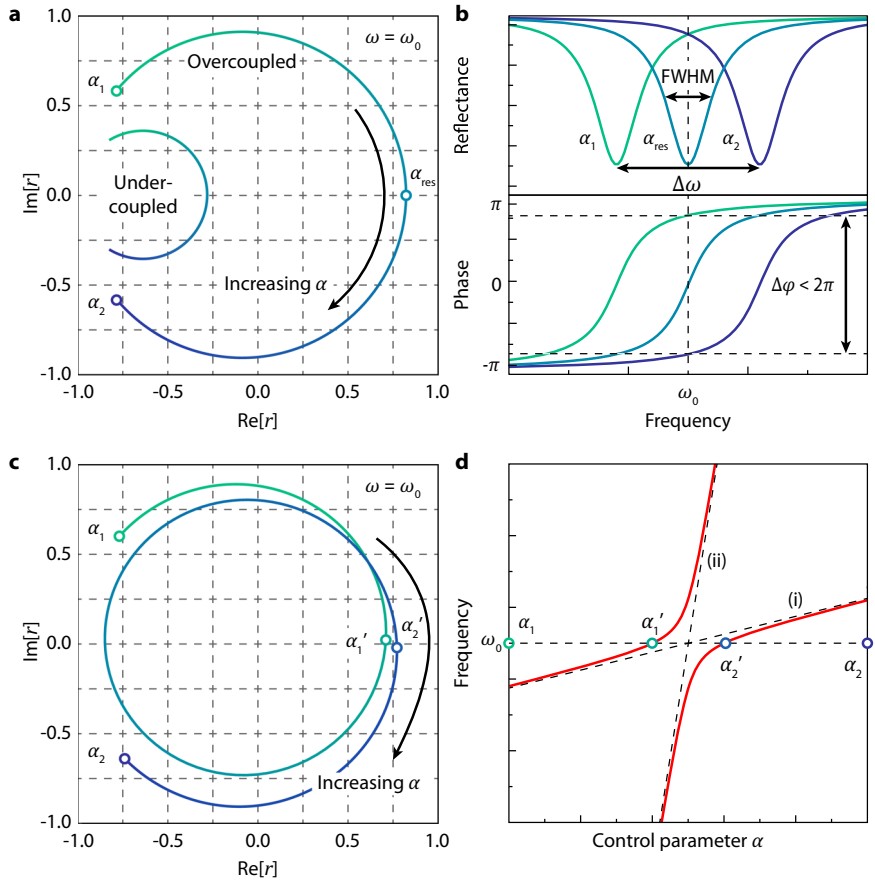

**Fig. 1 Problems in dynamic phase modulation and the usage of avoided crossing as a solution. a** The trajectories of the complex reflection coefficient for under-coupled and over-coupled resonances at the operating frequency $\omega_0$ as the single control parameter $\alpha$ is varied (color change from green to blue). Over-coupling is critical for modulating phase with a large coverage, as well as maintaining uniform amplitude. Over-coupling alone however cannot close the $0-2\pi$ loop. **b** The reflectance (top) and the phase (bottom) distributions of a single resonance as the control parameter is varied from $\alpha_1$ through $\alpha_{res}$ to $\alpha_2$. The trade-off due to the interdependence between the resonance shift and the spectral width results in a subpar figure of merit $\Delta\omega$/FWHM (full width at half maximum) insufficient for a complete $0-2\pi$ range. $\alpha_{res}$ corresponds to the control parameter value that results in the resonance frequency coinciding with $\omega_0$. **c** The trajectory of the complex reflection coefficient when an avoided crossing is used. Use of an avoided crossing circumvents the trade-off issue, closes the phase loop and modulates phase with uniform amplitude with an upper bound of $4\pi$. **d** The mode dynamics of the avoided crossing. The original resonance frequencies without the avoided crossing are shown with dashed lines, where resonance (i) is over-coupled and spectrally narrow while resonance (ii) has high resonance frequency tunability. The resonance frequencies of the coupled modes after the avoided crossing occurs are shown with solid red lines. The horizontal dashed line represents the operating frequency $\omega_0$. $\alpha_1'$ and $\alpha_2'$ are the values of the control parameter where the frequency of the coupled resonances coincide with $\omega_0$.

When $\triangle\epsilon_{tune}$ is increased by adding more charge carriers, the conductivity, and therefore $\mathrm{Im}[\epsilon_{tune}]$, also increases due to the higher carrier density[27,28]. This trade-off makes it technically difficult to use carrier injection to sufficiently maximize the figure of merit for covering the $0-2\pi$ range. Phase-change materials such as $VO_2$ and chalcogenide glasses provide dramatic dynamic modifications in $\triangle\epsilon$, but these materials tend to have significant $\mathrm{Im}[\epsilon]$, which limits the optical phase tunability[21,22]. Other tuning methods based on thermo-optic effects and liquid crystals may offer large $|\triangle\omega/\mathrm{FWHM}|$ as they are not inherently susceptible to the aforementioned issues[23,24]. However, these methods are limited in their local tunability, have slow operation speeds, and/ or require extreme temperatures, rendering them impractical for general applications.

To circumvent the trade-off issue described above and maintain uniform amplitude as shown in Fig. 1c, we utilize an avoided crossing between two resonances: an over-coupled, spectrally narrow resonance and a resonance with a high resonance frequency tunability. The two resonances must have overlapping field profiles, resulting in a non-zero coupling and

therefore exhibiting finite avoided crossing as exemplified in Fig. 1d. From the previous analysis, it is evident that spectrally narrow modes tend to have a large mode volume and therefore generally have small resonance shifts (small numerator over a large denominator in both Eqs. 1 and 2), while highly tunable modes tend to be spectrally broad. When coupled, the avoided crossing will then serve to hybridize the resonances, resulting in two modes with the combined features of being over-coupled and spectrally narrow, and having a large resonance shift. This grants three effects. First, the modes momentarily gain a superior $\triangle\omega/\mathrm{FWHM}$ as the avoidance between two modes provides an extra frequency shift, which is unavailable to conventional modes susceptible to the trade-off. Second, if the original spectrally narrow resonances were highly over-coupled, the hybrid modes would still be highly over-coupled near the anticrossing point and the phase modulation will possess nearly uniform amplitude as shown in Fig. 1c. Finally, because these modes sweep across the operating frequency subsequently one after the other, the first $0-2\pi$ phase loop is closed as the second

mode initiates the secondary phase loop. This renders an upper bound of $4\pi$ in phase modulation.

**A proof-of-concept metasurface.** The phenomenon of an avoided crossing between resonances is a general one, and a wide range of different experimental platforms hosting a narrow, over-coupled resonance and a resonance with high spectral tunability can be adopted to exhibit the aforementioned dynamic phase tuning scheme. These include plasmonic nanocavities[29] and molecules[30] with their myriad of plasmonic modes, and dielectric metasurfaces with various multipole excitations[31]. To validate the efficacy of the dynamic tuning scheme, however, we opted for a system that is easily experimentally accessible and designed a proof-of-concept reflective metasurface that supports dielectric-based qBICs and graphene plasmon resonances. Bound states in the continuum (BICs) are localized resonances whose eigenvalues are embedded within the continuous eigenvalue spectrum of radiating modes[32]. BICs turn into qBICs by detuning a certain system parameter slightly from a given value, which allows bound states to couple to the continuum, with the radiative loss becoming finite and controllable through said parameter. If dielectric resonances are used as a basis for the qBICs, the low dissipative loss stemming from the nature of dielectrics, along with the controllable radiative loss that can be tuned larger than the dissipative loss to make an over-coupled system, render dielectric qBICs an ideal candidate for the narrow, over-coupled resonance described above. Graphene plasmons, on the other hand, are excitations that couple electromagnetic waves and free electrons within the graphene. Known for their immense electric field concentration capabilities, small mode volume due to the material's 2D nature, and substantial tunability[27], graphene plasmonic resonances are good candidates for the second resonance with the high-frequency tunability (large numerator and small denominator in Eq. 1). Our final metasurface design is shown in Fig. 2a, with graphene nanoribbons situated between the Si pillars of a dimerized-high-contrast-grating, which hosts dielectric qBIC resonances[33] arising from the Brillouin zone-folding effect due to dimerization. This zone-folding effect for TM polarization is illustrated in Fig. 2b, c. The dark modes that were originally at the edge of the Brillouin zone outside the free-space light cones get folded into the interior of the light cones, allowing for free-space excitation. Because these modes are under the 1st order diffraction line, only the normally incident 0th-order modes need to be considered. The graphene plasmons are excited on the graphene ribbons that exist between the Si pillars, and there is substantial mode overlap between the qBIC and

graphene plasmonic resonances, which leads to strong coupling. Finally, a back reflector is used to double the $0-\pi$ phase range of single resonances as light reinteracts with the metasurface.

To make the qBIC highly over-coupled and therefore achieve uniform amplitude, we chose system parameters that yield a high radiative loss over the dissipative loss. While the dissipative loss is largely dictated by the material properties of graphene (e.g., interband and intraband transitions) and is thus difficult to control, the dominant dependence of the radiative loss on the system geometry allowed for the design of such. This is because the Si pillar width $w$, height $h$, and the degree of dimerizing perturbation $\delta$ tune the radiative coupling by affecting the strength of the dipole excitation between the dimerized pillars, whereas the substrate thickness $d$ controls the Fabry-Perot resonance between the Si pillars and the back reflector. Here, $\delta$ is defined as the lateral shift of the Si pillars normalized by the half-period $\Lambda/2$. The dependence of the loss rates on the system parameters can be found in Supplementary Note 1. The exact geometry values that rendered uniform amplitude as well as maximal phase modulation were obtained through numerical optimization (See Supplementary Note 2 for the detailed method). The optimized values of the geometric parameters are: $\Lambda = 5320$ nm, $h = 166$ nm, $d = 640$ nm, $w = 2482$ nm, and $\delta = 2.66$ %. The refractive indices for the Si pillars and the substrate are set to $n_{Si} = 3.42$, and $n_{substrate} = 2$, respectively. The back reflector was assumed to be PEC and the conductivity of the graphene was calculated with the Kubo formula at room temperature.

Figure 3 summarizes the performance of the metasurface as a reflective phase modulator for normally incident TM plane waves. The top row (Fig. 3a–c) shows the full-wave simulation results for the structure in the inset of Fig. 3a, where an unpatterned graphene sheet is placed underneath the Si pillars throughout the whole unit cell. In this configuration, even though the graphene sheet supports plasmons excited by the edges of the Si pillars, the plasmons simply propagate along with the continuous graphene sheet and dissipate without forming a significant resonance (electric field profile in Fig. 4a). Therefore, only the qBIC serves as the dominant resonance and no avoided crossing occurs. As the Fermi energy of graphene ($E_F$) is raised from 0 to 1 eV, the real part of graphene's permittivity decreases in the frequency region of interest, causing the qBIC resonance to blueshift as predicted by Eq. 1. The rate of the blueshift is 0.572 THz/eV, which is not significant enough for a full $2\pi$ phase shift ($\Delta\phi = 1.73\pi$) due to the trade-off discussed above. However, because the qBIC is highly over-coupled, the complex amplitude draws a near-

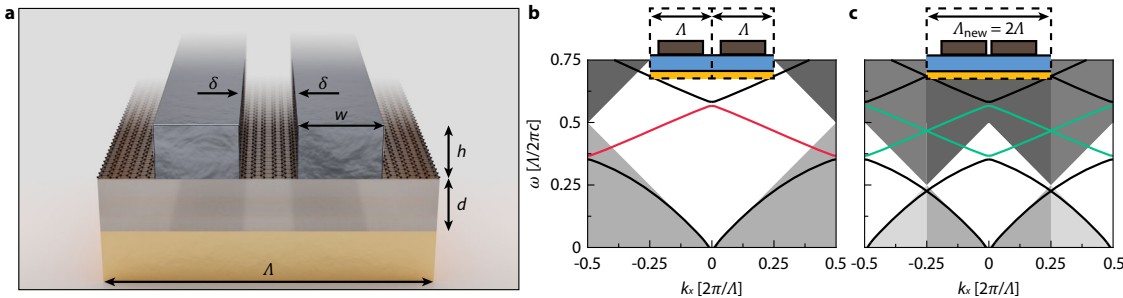

**Fig. 2 Schematic of the metasurface and the band structure of the gratings. a** Schematic of the metasurface. The graphene is patterned into nanoribbons between the Si pillars, forming graphene plasmons within a cavity. The Si pillars are grouped into dimers, resulting in a quasi-BIC from broken (doubled) periodicity of the Si grating. **b** The optical band structure before the dimerization of the pillars. The red line indicates the antibonding mode we will be utilizing. **c** The band structure after the dimerization of the pillars. Periodicity doubling causes zone folding, bringing in the modes that were previously inaccessible to free-space light into the light cone. The green line indicates the qBIC (antibonding mode). The white diamond region represents the 0th-order domain.

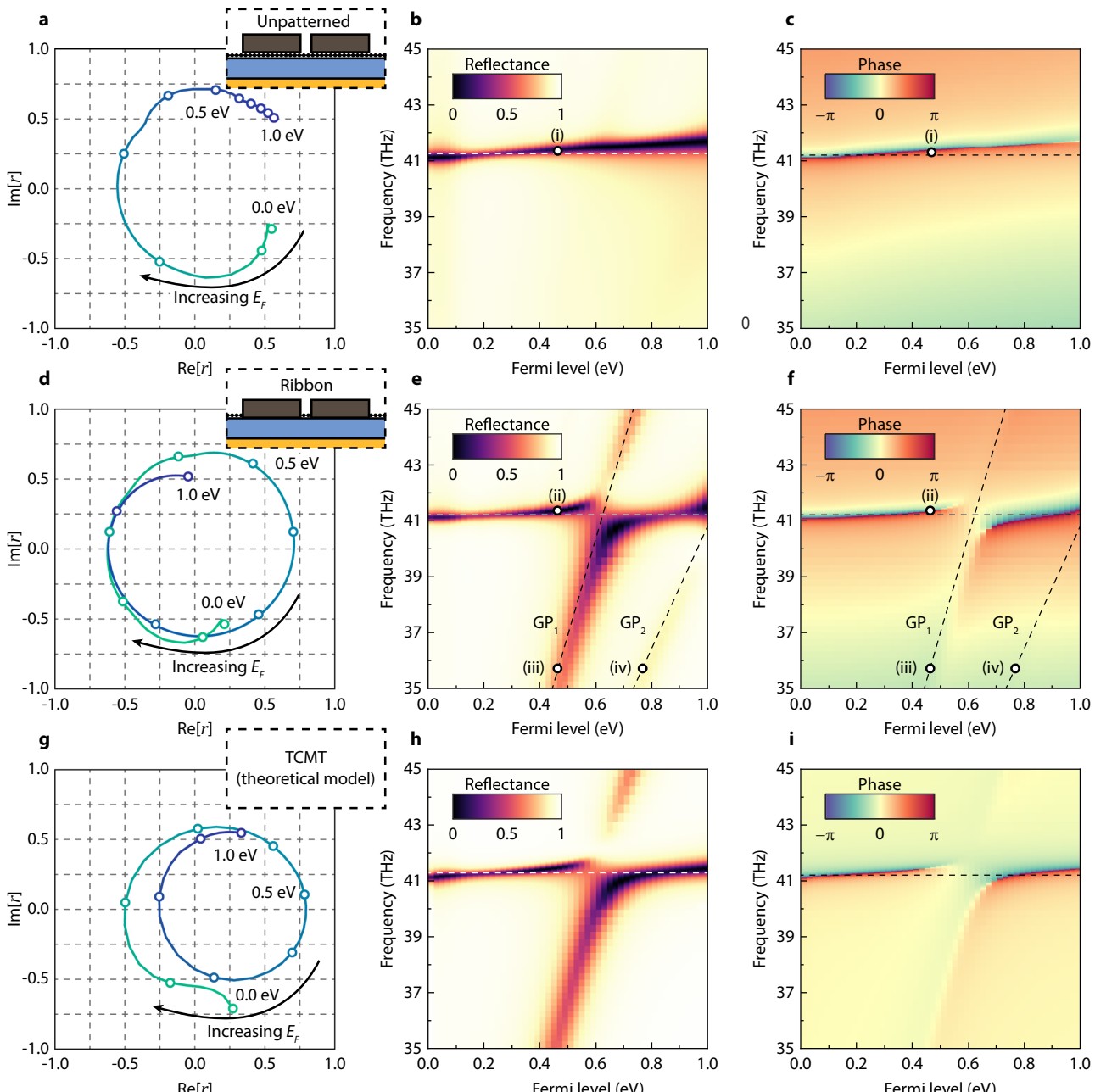

**Fig. 3 Phase modulation and the dynamics involved.** Full-wave calculations of the dependence of reflected light on $E_F$ for a dimerized Si grating on a **a**–**c** continuous sheet of graphene and **d**–**f** separated by parallel graphene nanoribbons of widths 36.7 nm (center) and 319.7 nm (side) as shown in the device structures schematically illustrated in the insets. **g**–**i** illustrate the results of the TCMT model described in the text. Panels **a**, **d**, **g**, show the complex reflection coefficients at 41.27, 41.20, and 41.27 THz, respectively, as $E_F$ is varied from 0 to 1 eV. Open circles throughout the curves indicate Fermi levels spaced out by 0.1 eV, starting from 0 eV. **b**, **e**, **h**, and **c**, **f**, **i**, plot the frequency dependence of the reflected intensity and phase, respectively.

uniform amplitude distribution. The detailed dependence of reflectance $R$ and reflection phase $\phi$ spectra on $E_F$ are present in Fig. 3b, c, respectively. The widening of the reflectance linewidth for $E_F < \hbar\omega/2 \sim 0.085$ eV in Fig. 3b originates from the graphene interband transition losses. The broadening of the reflection dip at high $E_F$ originates from the increased intraband conductivity of graphene[27].

On the other hand, Fig. 3d–f exhibit the full-wave simulation results for the metasurface shown in the inset of Fig. 3d where graphene is patterned into nanoribbons in between the Si pillars. In this case, the excited graphene plasmons bounce back and forth between the edges of the nanoribbons, forming a

strong resonance. The avoided crossing between the qBIC and the graphene plasmon resonance enables an unprecedented $3\pi$ phase revolution as well as exhibiting uniform amplitude of ~0.65 as seen in Fig. 3d. The avoided crossing between two resonances can be described by the following effective Hamiltonian

$$H = \begin{bmatrix} \omega_1 & u \\ u & \omega_2 \end{bmatrix} + i \begin{bmatrix} \gamma_{1d} + \gamma_{1r} & \sqrt{\gamma_{1r}\gamma_{2r}} \\ \sqrt{\gamma_{1r}\gamma_{2r}} & \gamma_{2d} + \gamma_{2r} \end{bmatrix}. \quad (4)$$

Here the $\omega_{1,2}$ are the resonance energies of the qBIC and the graphene plasmons before the interaction between them,

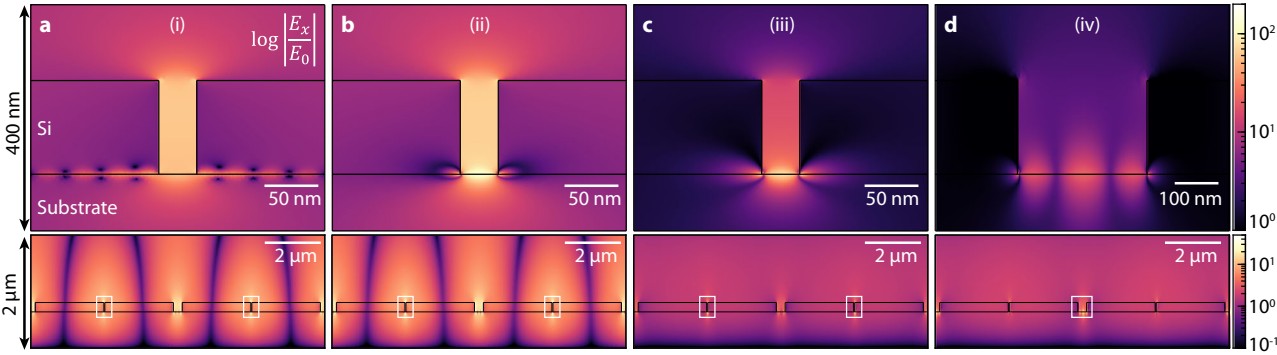

**Fig. 4 The electric field profiles of the modes.** The $E_x$ field profiles for the points (i), (ii), (iii), and (iv) in Fig. 3. Pictures on the top show close-up views of the sections enclosed by white rectangles in the bottom pictures. **a** The electric field of point (i) showing a qBIC with a weak off-resonance graphene plasmon excitation decaying underneath the pillars. **b** A similar qBIC mode of point (ii), with a localized off-resonance graphene plasmon excited in between the pillars on the graphene ribbon. **c** The graphene plasmon mode (GP$_1$) of point (iii) at the center of the unit cell. **d** The field of the secondary graphene plasmon (GP$_2$) mode excited on the sides of the unit cell at point (iv).

respectively. The coupling $u$ between the modes is proportional to their mode field overlap, and $\gamma_{1d,2d}$ and $\gamma_{1r,2r}$ are the dissipative and radiative energy loss rates of each mode, respectively. Diagonalizing the Hamitonian gives us the eigensolutions of $\omega_{\pm} = [\omega_1 + \omega_2 + i(\gamma_1 + \gamma_2)]/2 \pm \beta$, where $\beta^2 = (\omega_1 - \omega_2 + i(\gamma_1 - \gamma_2))^2/4 + (u + i\sqrt{\gamma_{1r}\gamma_{2r}})^2$ and $\gamma_{1,2} = \gamma_{1d,2d} + \gamma_{1r,2r}$. As discussed earlier, the sharp qBIC resonance $\omega_1(E_F) \propto E_F$ slowly blueshifts as $E_F$ increases. The electric field distribution of the qBIC mode is shown in Fig. 4b, exhibiting a field concentration in between the two pillars (antibonding mode). The graphene plasmon resonance, on the other hand, shifts much more drastically due to its small mode volume, and at a fixed resonator size is known to follow $\omega_2(E_F) \propto \sqrt{E_F}$[27], indicated as the dashed black curves labeled as GP$_1$ in Fig. 3e, f. The electric field distribution of GP$_1$ mode, plotted in Fig. 4c, shows a strong field localization near the graphene nanoribbons between the Si pillars. We note that the considerable field overlap between the qBIC and GP$_1$ modes ensures an avoided crossing between the two with a significant energy splitting $\beta$ ~2 THz near the crossing point. The spectral line cuts of the two resonances exhibiting the avoided crossing are shown in Supplementary Fig. 3 in Supplementary Information. Finally, there also exists a secondary graphene plasmon resonance (GP$_2$) taking place at the graphene ribbons on the edges of the unit cell where the gap between the two pillars is much wider due to the dimerization. The field profile of GP$_2$ is shown in Fig. 4d. The merging of the GP$_2$ with the qBIC mode around $E_F = 1$ eV results in the qBIC mode having a widened linewidth.

To gain deeper insights into the modulation behavior, we describe an analytical two-resonance model for the complex amplitude $r$ based on TCMT (see Supplementary Note 3 for the detailed formulation of the model). Here, the properties of each resonance are characterized by the resonance energy $\omega$ and dissipative ($\gamma_d$) and radiative ($\gamma_r$) loss rates. The coupling between the modes is described by the coupling constant $u$. The resulting complex amplitude revolution, reflectance, and phase are shown in Fig. 3g–i, respectively, which are similar to the full-field simulation results of Fig. 3d–f with minor differences, but still yields the $3\pi$ phase revolution in the complex reflectivity space. In both cases, qBIC exhibits an abrupt phase change while the graphene plasmon resonance (GP$_1$) does not as shown in Fig. 3f and i. This is because the graphene plasmon mode has a high dissipative loss rate compared to its radiative coupling ($\gamma_{2d} > \gamma_{2r}$) and thus is under-coupled to the incident light as opposed to the qBIC mode that is

over-coupled. We also observe that the reflection dip associated with the upper branch mode $\omega_+$ disappears at $E_F \approx 0.6$ eV, where the condition $u(\gamma_{1r} - \gamma_{2r}) = \sqrt{\gamma_{1r}\gamma_{2r}}(\omega_1 - \omega_2)$ holds. This is a general phenomenon that occurs in a broad range of systems[34], and can be interpreted as the energy transfer between the two resonances precisely counterbalancing the energy loss from one of the modes, turning it into a stable eigenstate. The discrepancy between the TCMT model and the full-field simulations can be explained by the following three reasons. First, the parameters used for the TCMT model are not exact, but rather crude approximations of the actual functions that subtly change according to the Fermi level. Second, the TCMT model does not include the effect of GP$_2$, leading to the discrepancy in the high $E_F$ regime. The comparison between Fig. 3d, g indicates that the interaction between the GP$_2$ and the qBIC resonance near the Fermi level of ~1 eV results in a slight improvement in the reflection amplitude $E_F > 0.8$ eV. Third, the TCMT model assumes a fixed background phase, whereas the actual background phase gradually varies with respect to the frequency as shown in Fig. 3f, due to the Fabry-Perot effect between the back reflector and the Si pillars. A more detailed comparison between the TCMT model and simulation results is provided in Supplementary Note 4, along with a discussion on the ideal TCMT parameters for phase modulation in Supplementary Note 6 and Supplementary Fig. 4 illustrating the dependence of the phase modulation performance on different TCMT parameters.

To demonstrate the viability of the metasurface for realistic levels of graphene quality currently achievable, as well as illustrate the ranges of amplitude achievable, Fig. 5a, b show complex reflection amplitude $E_F$ sweeps for different graphene mobilities in the metasurface structure with an unpatterned graphene sheet and graphene nanoribbons, respectively. For both cases, the tendency of the reflectivity circles to get bigger in amplitude is evident, due to the system having less dissipative loss and therefore becoming more over-coupled[13]. From a theoretical perspective, given that there are no material losses in the system, the amplitude will be near unity but there will still be small scattering losses from graphene plasmons reflecting off the Si pillars. Considering that there exist dielectrics and metals that have negligible losses in the infrared regime, the achievable amplitude range would be mainly determined by the loss in graphene. We note that the reflection amplitude can be as high as ~0.5 even at poor carrier mobility of $\mu_s = 500$ cm$^2$/V·s and can reach ~0.8 at $\mu_s = 2,000$ cm$^2$/V·s, which is achievable with commercial large-area graphene grown by chemical vapor deposition. With high-quality single-crystalline graphene having carrier mobilities exceeding 10,000 cm$^2$/V·s[35–37],

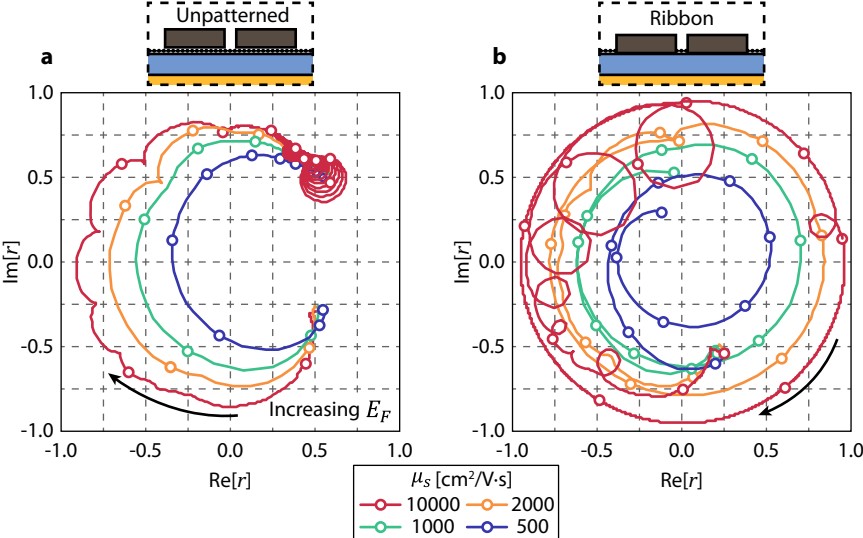

**Fig. 5 Phase modulation with different graphene mobility values. a** Complex reflection coefficient trajectories with different mobility values for the graphene sheet case. Full $2\pi$ phase modulation does not occur without the avoided crossing with graphene plasmons, despite the increasing mobilities and therefore the decreasing linewidths. **b** Complex reflection coefficient trajectories with different mobility values for the graphene ribbon case. Open circles throughout the curves in both A and B indicate Fermi levels spaced out by 0.1 eV, starting from 0 eV. The device structures are illustrated in the insets.

the maximum reflection amplitude of the device would reach near unity. However, the graphene sheet cases in Fig. 5a show that despite decreasing $\gamma_d$ and thus decreasing the FWHM, complete $2\pi$ phase modulation is not possible. On the other hand, the graphene nanoribbon cases in Fig. 5b utilizing the avoided crossing phenomenon all show ~$3\pi$ phase modulation capacities. Both cases of $\mu_s = 10,000 \mathrm{cm^2/V \cdot s}$ exhibit repeated round bulges (the graphene ribbon case showing more extreme circles). These can be attributed to high order graphene plasmonic modes that are present due to the extremely high mobility. The case of the graphene sheet has weaker graphene plasmon resonances due to the Si pillars acting as a partial reflection wall, whereas graphene ribbons have a near-complete reflection from the ribbon edges. Lastly, the graphene ribbon case shows reflection amplitude drops around $E_F = 1$ eV due to the absorption by the GP$_2$ mode (Fig. 3e).

The structural parameters of the nanoribbon device presented in Fig. 3c–e are designed to have maximum performance with the Fermi level tuning range of 0–1 eV. Such a large carrier density modulation has been experimentally demonstrated via ion-gel gating[38,39], but for the standard back gating scheme the tuning range of $E_F$ is bounded by the dielectric strength of the insulating layer. For a normal dielectric layer such as SiO$_2$ or SiN$_x$, an experimentally feasible $E_F$ tuning range is around $E_F = 0$–0.6 eV[40,41]. With this limited $E_F$ tuning range, the device presented in Fig. 3c–e can only cover 282° phase range. The reason for this sub-$2\pi$ phase coverage is that the device is only able to utilize one resonance in the $E_F$ range since the avoided crossing between the graphene plasmon resonance and the qBIC mode occurs at the upper bound of the $E_F$ range ($E_F = 0.6$ eV). We note that, by altering the device parameters to have avoided crossing at a lower Fermi energy, it is still possible to achieve full $2\pi$ phase coverage for the limited $E_F$ tuning range of 0–0.6 eV with a marginal decrease in reflection amplitude as shown in Supplementary Fig. 5 in the Supplementary Information.

## Discussion
We present a solution to the fundamental trade-off problem of dynamic metasurfaces, where broad (narrow) resonances tune more (less) leading to insufficient phase tunability. We show that at the

avoided crossing point of two coupled resonators, the hybridized modes that subsequently cross the operating frequency possess the qualities of being over-coupled and highly dispersive, allowing for a greater than $2\pi$ phase shift, and that through optimization of the geometric parameters, the amplitude can be kept constant throughout the tuning range. The upper theoretical limit of phase tuning using this methodology is $4\pi$. As a proof-of-concept, we demonstrate a reflective metasurface that incorporates proven metasurface elements made from real materials—graphene nanoribbons and dimerized silicon gratings—and performs an over-$3\pi$ revolution in the complex reflectivity space with uniform amplitude. These results are achieved using only a single control parameter—the carrier density of the graphene. Although this specific metasurface operates in the mid-infrared frequency regime, an avoided crossing is in principle applicable to any frequency regime with appropriate resonances. We anticipate that this methodology will provide a go-to formula for full $2\pi$ dynamic phase modulation schemes, and open a promising doorway for active metasurface design.

## Methods
For the numerical optimization of our metasurfaces, we employed Reticolo RCWA (rigorous coupled-wave analysis) software and the gradient-free algorithm BOBYQA (bound optimization by quadratic approximation) (See Supplementary Information for details), to obtain the exact geometric parameters for maximal phase modulation. In the BOBYQA optimization, the FoM for the optimization was defined as the area enclosed by the complex reflectivity curve drawn on the complex plane as the graphene Fermi level was tuned at a single target frequency. This definition facilitated the simultaneous optimization of both the reflection amplitude and the phase coverage. In the RCWA simulation, for the structure with the graphene sheet, the Fourier order needed for RCWA to reproduce similar reflectance spectra to FEM results was only 100, for which evaluating 51 Fermi energy points took less than 5 s on our server computer (Intel E5-2680v4). For the structure with the graphene ribbon, the required Fourier order was about 600, which took roughly 200 times longer than that of the case with 100 Fourier orders. The reason for such a long computation time stems from the extremely narrow width of the graphene nanoribbons compared to the structure period. In order to bypass this long computation time, we first optimized geometric structural parameters for a maximum FoM at low Fourier orders, and then set the obtained geometric parameters as an initial point for the next optimization procedure with higher Fourier orders. The optimized metasurfaces' optical responses simulated by the RCWA were compared to that of the commercial FEM tool (COMSOL Multiphysics), which showed excellent agreement.

## Data availability

All data needed to evaluate the conclusions in this study are presented in the manuscript and in the Supplementary Information.

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

## Acknowledgements

This work was supported by the Samsung Research Funding & Incubation Center of Samsung Electronics under Project Number SRFC-IT1702-14.

## Author contributions

J.Y.K., J.P., and M.S.J. conceived the idea. G.R.H. and V.W.B. contributed to the further development of the idea. J.Y.K. and M.S.J. conducted theoretical analysis. J.P. conducted device optimizations. J.P., J.T.H., and S.K. performed electromagnetic simulations. J.T.H. directed visualization. J.Y.K., J.P., and M.S.J. analyzed the data. J.Y.K., J.P., V.W.B., and M.S.J. wrote the manuscript. V.W.B. and M.S.J. supervised the project.

## Competing interests

The authors declare no competing interests.
