## [Peer Review File · Nature Communications]

Full 2π Tunable Phase Modulation Using Avoided Crossing of ResonancesREVIEWER COMMENTS

Reviewer #1 (Remarks to the Author):

The authors present a highly novel solution to a fundamental problem in dynamic metasurfaces, relating to the issue of maximal active tunability and high quality resonances. They uncovered an interesting result, that at the avoided crossing between two coupled resonators, the hybridized modes possesses the qualities of both being over coupled and highly dispersive, allowing for a greater than 2π phase shift, with theoretical upper bound being 4π . Through proper designs, using graphene nanoribbons and dimerized silicon gratings, they are able to demo a performance over 3π modulation with uniform reflection amplitude through just tuning the carrier density of graphene. To the best of my knowledge, the proposal of utilizing the avoided crossing of coupled resonators in reflectarray is new and creative. The topic is certainly of great interest to the community and timely, and the manuscript is well written and results scientifically sound. I recommend it's publication in Nature Comm with the following suggestions,

1. The range of graphene doping in some of the examples are rather high. I think experimentally a doping of 0.4-0.5 eV would probably be the upper limit for standard back gating scheme. The authors should discuss what can be done on the design aspect to achieve similar modulation range but at smaller doping.
2. Would one yield any benefit in the designs by utilizing dissimilar graphene ribbons within the meta molecule supercell?
3. The authors should discuss or emphasize more the importance of the choice of using Si pillars and graphene, particularly in terms of their contrasting linewidth. Can one learn from here that greater contrasting linewidth is more beneficial for the performance?

Reviewer #2 (Remarks to the Author):

The authors proposed a means of achieving $> 2\pi$ (theoretically up to 4π) dynamic phase modulation with the use of dimer structure to be used in metasurface applications.

The dimer structure provides a broadband widely tunable mode plus a narrowband mode, when hybridized, provide total of $2 \times 2\pi = 4\pi$ theoretical modulation capability.

In result, the authors obtain EIT-like narrowband reflection, for its wideband tuning supported by the index tuning of wideband mode.

The other idea implemented to keep the reflection amplitude constant during the tuning process, was the over-coupling to radiation mode, which screens the sensitivity of unloaded Q factor.

Authors justified/verified their proposal/idea in view of variational theorem and analytical CMT, while carrying out detailed numerical analysis.

While it was not clear whether the result includes experimental verification until reading the manuscript throughout to the end, the numerical analysis using the experimentally accessible template

(of graphene nanoribbons and dimer silicon gratings), strongly supports their proposal with the observed phase evolutions over complex reflectivity plane and dispersion curves.

Overall, the proposed idea and underlying physics is clear, and the manuscript reads relatively well. Considering that the authors provided a means of overcoming the present hurdle

in the design of meta-atom for metasurface application (dynamic phase $> 2\pi$, at constant reflectivity), this reviewer supports the publication of this manuscript without the need of experimental verification.

Meanwhile, to better the manuscript in terms of readability and completeness, this AE suggests following modifications in their revision.

1. It is suggested to make it clear in the abstract and introduction, that the idea has been justified analytically and verified numerically in the experimentally accessible platform.

2. For general readership, it is suggested to make it clear somewhere in the introduction, that avoided crossing is equivalent to anti-crossing; such as avoided crossing (a.k.a. anti-crossing).

3. The part of “conventional limits in phase modulation” (line 73-135) is little bit lengthy and somewhat redundant.

Eq.1 and Eq.2 well-known in community, authors may consider using those equation in the text (without number) while keeping Eq.3 and concentrating its significance.

4. For the generality of the proposed approach, it would be great if authors could list some other types of experimental template.

E.g., state some other possible templates for the generality, around line 138, and then write.. “here we focus.. the present example.. especially advantageous... experimentally easily accessible.. “

5. Basically, the tuning is achieved from $w_2(EF)$ - which is evident from the equation. Is it possible to qualitatively interpret this result in view of EIT, where the background broadband mode is tuned by EF?

6. Before conclusion, in the section of EF sweeping (line245-260), this reviewer expected to see 3π phase change in the text (shown in Fig.5, but hard to read the graph).

To remove the possible confusion, it is suggested to clearly write $\sim 3\pi$ phase evolutions are obtained for various u_s values, for the change of $EF=0$ to $EF(u_s)$. Maybe between line 251 and 252?

7. In Fig.3, it is recommended to overlay the reflectance and phase curves along the line of evolution (the present color scale is hard to quantify)

For example, authors may consider reflectance/phase plot (curve) overlaid just inside (or outside) the x axis (Fig. 3b,e,h/3c,f,i), while little more emphasizing the line of tracing (frequency).

This will make it easier for the readers to follow/compare the phase evolution and the reflectance spectra, for the un-patterned / ribbon / TCMT analysis.

8. This reviewer may have missed, but it will be better if authors clarify the ranges of amplitude which could be achieved, in terms of theoretical, and practical viewpoint.

We thank the Reviewers for their valuable comments. We have carefully revised the manuscript in light of all the received questions and comments that are marked in blue. Our point-by-point responses to them are marked in black.

Reviewer 1

The authors present a highly novel solution to a fundamental problem in dynamic metasurfaces, relating to the issue of maximal active tunability and high quality resonances. They uncovered an interesting result, that at the avoided crossing between two coupled resonators, the hybridized modes possesses the qualities of both being over coupled and highly dispersive, allowing for a greater than 2π phase shift, with theoretical upper bound being 4π . Through proper designs, using graphene nanoribbons and dimerized silicon gratings, they are able to demo a performance over 3π modulation with uniform reflection amplitude through just tuning the carrier density of graphene. To the best of my knowledge, the proposal of utilizing the avoided crossing of coupled resonators in reflectarray is new and creative. The topic is certainly of great interest to the community and timely, and the manuscript is well written and results scientifically sound. I recommend its publication in Nature Comm with the following suggestions.

We thank the Reviewer 1 for the kind consideration of our work and valuable comments. We address each of the Reviewer's questions below.

1. The range of graphene doping in some of the examples are rather high. I think experimentally a doping of 0.4-0.5 eV would probably be the upper limit for standard back gating scheme. The authors should discuss what can be done on the design aspect to achieve similar modulation range but at smaller doping.

As the Reviewer pointed out, it is challenging to experimentally tune the Fermi level of graphene up to 1 eV. Such a large carrier density modulation has been demonstrated to be possible via ion-gel gating (see for example, *Nanoscale* **7**, 19493-19500 (2015) or *Nature* **471**, 617-620 (2011)), but for the standard back gating scheme the tuning range of E_F is bounded by the dielectric strength of the insulating layer. For a normal dielectric layer such as SiO_2 or SiN_x , experimentally feasible E_F tuning range is around $E_F=0-0.6\text{eV}$. (*Nano Lett.* **18**, 971-979 (2018) and *Nat. Commun.* **7**, 1-8 (2016))

The device presented in the main paper can only cover 282° for $E_F = 0-0.6\text{eV}$ as shown in Fig. 3d. The reason for this sub- 2π phase coverage is that the device is utilizing only one resonance in this E_F range since the avoided crossing between the graphene plasmon resonance and qBIC mode occurs at the upper bound of the E_F range ($E_F = 0.6\text{eV}$). We would like to note again that to achieve full 2π coverage, it is necessary to have a double crossover between the operating frequency line and the two hybrid modes. Therefore, to have a better phase coverage with a limited E_F tuning range, the device parameters should be altered to have the avoided crossing at a lower Fermi level. This can be done by either narrowing the center graphene width to obtain the GP resonance at a lower Fermi energy or by increasing the meta-atom period to lower the frequency of the qBIC resonance. The other design parameters, which determines the radiative and dissipative decay rates of the resonances, also need to be fine-tuned to

achieve a nearly uniform reflection amplitude.

To test this possibility, we have conducted an additional design optimization for 0-0.6eV Fermi level range as shown in Fig. R1. Compared to the structure presented in the main paper (blue line), the low- E_F -optimized device (red line) has a 15.6% longer period Λ but the gap between the Si pillars are nearly the same for the two structures. As shown in Fig. R1a, the reflection curve of the low- E_F -optimized device can cover full 2π phase change while its magnitude is slightly lower than the curve in Fig. 3d. Note that, as shown in Fig. R1b and R1c, the avoided crossing occurs at around $E_F = 0.5$ eV, which is included in the 0-0.6 eV tuning range.

Figure R1. **a**, The complex reflection amplitude plots for the structure presented in the main paper and a structure optimized for a limited E_F range of $E_F = 0-0.6$ eV. The parameters are: period $\Lambda = 6155$ nm, $h = 61$ nm, $d = 896$ nm, $w = 888$ nm, $\delta = 34.9\%$, center gap = 38.7 nm, side gap = 4340 nm, and operating frequency = 37.64 THz. **b**, **c**, The reflectance and the phase color plot, respectively, for the new structure. The operating frequency is shown as the dotted line.

We have added the discussion on the limited E_F tuning range in **Supplementary Section S7** and also added an additional paragraph right before the **conclusion** to illustrate this point as follows:

“The structural parameters of the nanoribbon device presented in Fig. 3c-e are designed to have maximum performance with the Fermi level tuning range of 0-1 eV. Such a large carrier density modulation has been experimentally demonstrated via ion-gel gating (38, 39), but for the standard back gating scheme the tuning range of E_F is bounded by the dielectric strength of the insulating layer. For a normal dielectric layer such as SiO_2 or SiN_x , experimentally feasible E_F tuning range is around $E_F = 0-0.6$ eV. (40,41). With this limited E_F tuning range, the device presented in Fig. 3c-e can only cover 282° phase range. The reason for this sub- 2π phase coverage is that the device is only able to utilize one resonance in the E_F range since the avoided crossing between the graphene plasmon resonance and the qBIC mode occurs at the upper bound of the E_F range ($E_F = 0.6$ eV). We note that, by altering the device parameters to have the avoided crossing at a lower Fermi energy, it is still possible to achieve full 2π phase coverage for the limited E_F tuning range of 0-0.6 eV with a marginal decrease in reflection amplitude as shown in Fig S5 in the Supplementary Materials.”

2. Would one yield any benefit in the designs by utilizing dissimilar graphene ribbons within the meta molecule supercell?

The reason the graphene ribbon widths are different in our structure design stems from the fact that the Si pillars have been symmetry-broken, meaning they have been dimerized (each pair of pillars have been brought closer together) to induced a quasi-BIC from a perfect BIC. However, because the avoided crossing occurs between this quasi-BIC and the graphene plasmon resonance that exists on the graphene ribbon placed between each dimerized pair, the “side graphene ribbons” that are placed between different dimerized pairs may seem unnecessary. Although this is a valid point, we would like to mention that the side graphene ribbons also host a graphene plasmon resonance of their own (referred to as the GP₂ mode in the main paper), which play a minor role in interacting with the quasi-BIC resonance near the Fermi level of ~1eV. This brings a slight improvement on the dynamic tuning result, as evidenced by the comparison between the device results Fig. 3d,e,f and the theoretical TCMT results Fig. 3g,h,i which ignore the GP₂ mode. To make this point clear, we have added the following sentence on **page 12** of the main paper (the added text is underlined):

“...Second, the TCMT model does not include the effect of GP₂, leading to the discrepancy at high E_F regime. The comparison between Fig. 3d and Fig. 3g implies that the interaction between the GP₂ and the qBIC resonance at $E_F \sim 1$ eV results in a slight improvement on the reflection amplitude at $E_F > 0.8$ eV.”

We can also consider the possibility of altering the width of the side graphene ribbons while keeping the width of the wider gap between the Si pillars unchanged, although it could be experimentally challenging to implement. This can offer an interesting tuning knob that can fine-tune the reflectivity response around the crossing point between the qBIC and the GP₂ resonance to further enhance the phase-tuning performance. However, we anticipate that the avoided crossing between the qBIC and the GP₂ would not be significant enough to result in a proper double anti-crossing even in this case. This is because the mode overlap between the qBIC mode and the GP₂ mode is not as great as the mode overlap between the main graphene plasmon mode GP₁ and the qBIC mode, as evidenced by the mode electric field profiles shown in Fig. 4 of the manuscript. Nevertheless, there still exists a possibility to induce a double anti-crossing by putting multiple graphene nanoribbons within the same Si gap to enhance the coupling, which in principle allows for multiple phase revolutions within the same E_F tuning range.

3. The authors should discuss or emphasize more the importance of the choice of using Si pillars and graphene, particularly in terms of their contrasting linewidth. Can one learn from here that greater contrasting linewidth is more beneficial for the performance?

(i) Performance vs. linewidth contrast

We thank the Reviewer for the opportunity to clarify the relationship between the phase tuning performance and the linewidths of the two resonances. First of all, we would like to point out that the aim of the solution is not to use two resonances with contrasting linewidths, but instead to use a spectrally ‘narrow’, over-coupled resonance and a resonance with a ‘high resonance frequency shift/tunability’ (high $\Delta\omega$) as the control parameter is changed. It seems like the two resonances have contrasting linewidths because the resonance with a large $\Delta\omega$ tends to have a large linewidth due to the nature of the trade-off described in the main text (i.e. large tunability leads to a large linewidth). Therefore, instead of focusing on the linewidth contrast, it would be more appropriate to compare the linewidth of the spectrally narrow resonance and $\Delta\omega$ of the highly-tunable resonance.

The Reviewer's question is then interpreted as "Is it better performance-wise for the spectrally narrow resonance to have the smallest FWHM as possible and for the second resonance to have the greatest $\Delta\omega$ as possible?" The answer is more complex than a simple yes. To illustrate, we have added an additional figure **Fig. S4** and the corresponding explanation in **Section S6** in the **Supplementary Information** and have modified the final sentence at the end of the second-to-last paragraph before the conclusion in the main text as follows (**page 12**, the added text is underlined):

"A more detailed comparison between the TCMT model and simulation results is provided in Supplementary Section 4, along with a discussion on the ideal TCMT parameters for phase modulation and in Supplementary Section 6 and Fig. S4 illustrating the dependence of the phase modulation performance on different TCMT parameters."

In general, the decreasing FWHM of the spectrally narrow resonance (denoted as Resonance 1 hereafter) with respect to $\Delta\omega$ of the second resonance (denoted as Resonance 2 hereafter) leads to a better device performance (See Fig. 1b main text). However, there exist a few additional requirements that must be satisfied simultaneously:

- **Resonance 1 should be highly over-coupled ($\gamma_{1r} \gg \gamma_{1d}$):** The FWHM of Resonance 1 can be decomposed to $\text{FWHM}_1 = 2(\gamma_{1r} + \gamma_{1d})$, where γ_{1r} and γ_{1d} are the radiative and dissipative coupling rate of Resonance 1. Even if the FWHM_1 is very small, if Resonance 1 is under-coupled ($\gamma_{1r} < \gamma_{1d}$) the device will fail to cover the range $0-2\pi$ in phase modulation as shown in Fig. 1a and Fig. R2b. A decrease in γ_{1d} while γ_{1r} is kept constant, on the other hand, is beneficial not only because it decreases the FWHM_1 but also increases the ratio $p = \gamma_r/\gamma_d$, thereby increasing the reflection amplitude through $R = |r|^2 = 1 - 4p/(p + 1)^2$ and making it more uniform as shown in Fig. R2c. It is evident that the complex reflection amplitude draws a larger phase modulation circle while having a more uniform radius in Fig. R2c than the reference case in Fig. R2a.
- **γ_{1r} should not be too small compared to γ_{2d} :** Even if it's ensured that Resonance 1 stays highly over-coupled ($\gamma_{1r} \gg \gamma_{1d}$), decreasing γ_{1r} does not always bring performance enhancement. If γ_{1r} is much smaller than γ_{2d} of Resonance 2 in terms of their order of magnitudes ($\gamma_{1r} \ll \gamma_{2d}$), the phase modulation will fail to cover the range $0-2\pi$ as shown in Fig. R2d, a case where both γ_{1r} and γ_{1d} are decreased while keeping γ_{2d} the same as the reference. This is because the hybridized mode (with the characteristics of having both γ_{1r} and γ_{2d}) will be too under-coupled near the avoided crossing region and the phase modulation will fail to circle around the origin.
- **Resonance 2 should be highly tunable ($\Delta\omega_2 > \text{FWHM}_1$):** Having a greater $\Delta\omega_2$ will lead to a better phase modulation performance. This is because the hybridized modes will cross the operating frequency in greater speed as the control parameter (in our case the Fermi level) is varied. This is illustrated in Fig. R2e.

From the discussions above, we can combine the different cases and plot the ideal case of having large $\Delta\omega_2$ and small $\gamma_{1r}, \gamma_{1d}, \gamma_{2d}$ with $\gamma_{1r} \gg \gamma_{1d}$ & $\gamma_{1r} > \gamma_{2d}$, as shown in Fig. R2f.

Figure R2 (Figure S4). Dependence of the phase modulation performance with respect to different TCMT parameters shown in Table S1. a, The complex reflection amplitude for the reference TCMT parameter case. **b,** The under-coupled case where $\gamma_{1r} = \gamma_{1r}^0/30$ and $\gamma_{1d} = \gamma_{1d}^0$, with other parameters kept the same. The details are shown in Table R1 below. **c,** The case where $\gamma_{1r} = \gamma_{1r}^0$ and $\gamma_{1d} = \gamma_{1d}^0/30$. **d,** The case where $\gamma_{1r} = \gamma_{1r}^0/30$ and $\gamma_{1d} = \gamma_{1d}^0/30$ with $\gamma_{2d} = \gamma_{2d}^0$. **e,** The case where $\Delta\omega_2 = 3\Delta\omega_2^0$. **f,** The ideal case where $\gamma_{1r} = \gamma_{1r}^0$, $\gamma_{1d} = \gamma_{1d}^0/30$, $\gamma_{2d} = \gamma_{2d}^0/30$ and $\Delta\omega_2 = 3\Delta\omega_2^0$.

Case	$\Delta\omega_2$	γ_{1r}	γ_{1d}	γ_{2d}
Reference (Fig. R2A)	$\Delta\omega_2^0$	γ_{1r}^0	γ_{1d}^0	γ_{2d}^0
Undercoupled qBIC (Fig. R2B)	$\Delta\omega_2^0$	$\gamma_{1r}^0/30$	γ_{1d}^0	γ_{2d}^0
Small γ_{1d} (Fig. R2C)	$\Delta\omega_2^0$	γ_{1r}^0	$\gamma_{1d}^0/30$	γ_{2d}^0
Small γ_{1r} , Small γ_{1d} (Fig. R2D)	$\Delta\omega_2^0$	$\gamma_{1r}^0/30$	$\gamma_{1d}^0/30$	γ_{2d}^0
Large $\Delta\omega_2$ (Fig. R2E)	$3\Delta\omega_2^0$	γ_{1r}^0	γ_{1d}^0	γ_{2d}^0
Ideal (Fig. R2F)	$3\Delta\omega_2^0$	γ_{1r}^0	$\gamma_{1d}^0/30$	$\gamma_{2d}^0/30$

Table R1. The numerical values/functions of the TCMT parameters used in Fig. R2.

(ii) Importance of the choice of using Si pillars and graphene

According to the Reviewer's suggestion, to clearly emphasize the importance of the choice of using Si pillars and graphene, we have modified the paragraph starting with the introduction of qBICs under the **Proof-of-concept metasurface** section on **page 8** as follows:

“... BICs turn into qBICs by detuning a certain system parameter slightly from a given value, which allows bound states to couple to the continuum, with the radiative loss becoming finite and controllable through said parameter. If dielectric resonances are used as a basis for the qBIC, the low dissipative loss stemming from the nature of dielectrics, along with the controllable radiative loss that can be tuned larger than the dissipative loss to make an over-coupled system, render dielectric qBICs an ideal candidate for the narrow, over-coupled resonance described above. Graphene plasmons, on the other hand, are excitations that couple electromagnetic waves and free electrons within the graphene. Known for their immense electric field concentration capabilities, small mode volume due to the material's 2D nature, and their substantial tunability (27), graphene plasmonic resonances are good candidates for the second resonance with the high frequency tunability...”

Also, Si pillars and graphene plasmons are not the only unique choice for such a platform for an avoided crossing. For generality, we have also added the following at the beginning of the **Proof-of-concept metasurface** section:

“The phenomenon of an avoided crossing between resonances is a general one, and a wide range of different experimental platforms hosting a narrow, over-coupled resonance and a resonance with high spectral tunability can be adopted to exhibit the aforementioned dynamic phase tuning scheme. These include plasmonic nanocavities (29) and molecules (30) with their myriad of plasmonic modes, dielectric metasurfaces with various multipole excitations (31). To validate the efficacy of the dynamic tuning scheme, here we opted for a system that is easily experimentally accessible and designed a proof-of-concept reflective metasurface that supports quasi-bound states in the continuum (qBICs) and graphene plasmons...”

Reviewer 2

The authors proposed a means of achieving $> 2\pi$ (theoretically up to 4π) dynamic phase modulation with the use of dimer structure to be used in metasurface applications. The dimer structure provides a broadband widely tunable mode plus a narrowband mode, when hybridized, provide total of $2 \times 2\pi = 4\pi$ theoretical modulation capability. In result, the authors obtain EIT-like narrowband reflection, for its wideband tuning supported by the index tuning of wideband mode. The other idea implemented to keep the reflection amplitude constant during the tuning process, was the over-coupling to radiation mode, which screens the sensitivity of unloaded Q factor. Authors justified/verified their proposal/idea in view of variational theorem and analytical CMT, while carrying out detailed numerical analysis.

While it was not clear whether the result includes experimental verification until reading the manuscript throughout to the end, the numerical analysis using the experimentally accessible template (of graphene nanoribbons and dimer silicon gratings), strongly supports their proposal with the observed phase evolutions over complex reflectivity plane and dispersion curves.

Overall, the proposed idea and underlying physics is clear, and the manuscript reads relatively well. Considering that the authors provided a means of overcoming the present hurdle in the design of meta-atom for metasurface application (dynamic phase $> 2\pi$, at constant reflectivity), this reviewer supports the publication of this manuscript without the need of experimental verification.

Meanwhile, to better the manuscript in terms of readability and completeness, this AE suggests following modifications in their revision.

We thank the Reviewer 2 for the valuable feedback. We address each of the Reviewer's questions below.

1. It is suggested to make it clear in the abstract and introduction, that the idea has been justified analytically and verified numerically in the experimentally accessible platform.

According to the Reviewer's suggestion, we have made the following changes to the manuscript (the added text is underlined).

In the **abstract**: "... A proof-of-concept metasurface is justified analytically and verified numerically in an experimentally accessible platform using quasi-bound states in the continuum (qBICs) and graphene plasmon resonances ..."

In the **introduction**: "... Finally, we theoretically justify and numerically verify a proof-of-concept metasurface in an experimentally accessible platform utilizing quasi-bound states in the continuum (qBIC) and graphene plasmons..."

2. For general readership, it is suggested to make it clear somewhere in the introduction, that avoided crossing is equivalent to anti-crossing; such as avoided crossing (a.k.a. anti-crossing).

As the Reviewer suggested, we have added the parentheses '(also known as anti-crossing)' after the first use of the term 'avoided crossing' in the introduction on **page 4** of the manuscript.

3. The part of “conventional limits in phase modulation” (line 73-135) is little bit lengthy and somewhat redundant. Eq.1 and Eq.2 well-known in community, authors may consider using those equation in the text (without number) while keeping Eq.3 and concentrating its significance.

As the Reviewer commented, Eq.1 and Eq.2 may be relatively well-known in the nanophotonics community. However, we would like to point out that these equations are referred four times in the manuscript. Without the numberings, it would be difficult to point to these equations in a succinct manner. We therefore prefer to keep these equations numbered.

4. For the generality of the proposed approach, it would be great if authors could list some other types of experimental template. E.g., state some other possible templates for the generality, around line 138, and then write “here we focus... the present example... especially advantageous... experimentally easily accessible.”

We thank the Reviewer for the great suggestion. We have added the following sentences to the beginning of the section **A proof-of-concept metasurface**:

“The phenomenon of an avoided crossing between resonances is a general one, and a wide range of different experimental platforms hosting a narrow, over-coupled resonance and a resonance with high spectral tunability can be adopted to exhibit the aforementioned dynamic phase tuning scheme. These include plasmonic nanocavities (29) and molecules (30) with their myriad of plasmonic modes, dielectric metasurfaces with various multipole excitations (31). To validate the efficacy of the dynamic tuning scheme, here we opted for a system that is easily experimentally accessible and designed a proof-of-concept reflective metasurface that supports quasi-bound states in the continuum (qBICs) and graphene plasmons...”

5. Basically, the tuning is achieved from $w_2(EF)$ - which is evident from the equation. Is it possible to qualitatively interpret this result in view of EIT, where the background broadband mode is tuned by EF?

We thank the Reviewer for the astute comment. The answer to this question is both a yes and a no. The yes comes from the fact that our system involves two resonances/resonators that are coupled to one another, with one broad and one narrow ($\gamma_2 \ll \gamma_1$), and therefore one may set up the equation of the very general model of a pair of coupled resonators that is used frequently in the discussion of Fano resonances and EIT (Electromagnetically Induced Transparency). However, the no comes from the crucial difference between our system and systems that typically exhibit EIT: This is that in our system both resonances are coupled to the incident light (both are bright modes) whereas EIT systems involve only one resonance that is coupled to the incident light and the other that is not (a dark mode). Using the formalism of *Phys. Rev. Lett.* 101, 047401 (2008) EIT is described by the equation:

$$\begin{pmatrix} \tilde{a} \\ \tilde{b} \end{pmatrix} = - \begin{pmatrix} \delta + i\gamma_a & \kappa \\ \kappa & \delta + i\gamma_b \end{pmatrix}^{-1} \begin{pmatrix} g\tilde{E}_0 \\ 0 \end{pmatrix}$$

Where $|a\rangle = \tilde{a}(\omega)e^{i\omega t}$ and $|b\rangle = \tilde{b}(\omega)e^{i\omega t}$ are the radiative state and the dark state, respectively, $\delta = \omega - \omega_0$ is the frequency detuning from the resonant frequency ω_0 of both $|a\rangle$ and $|b\rangle$, $\gamma_{a,b}$ are the damping rates of $|a\rangle$ and $|b\rangle$ with $\gamma_b \ll \gamma_a \ll \omega_0$, κ is the coupling between the two states and g is the geometric parameter indicating how strong the state $|a\rangle$ couples to the incident light. Assuming zero detuning $\delta = 0$ and solving for the amplitude response of the radiative state we get $\tilde{a} = ig\tilde{E}_0/(\gamma_a + \kappa^2/\gamma_b)$. Now if the dark state's damping rate is many orders of magnitude smaller than

the coupling constant we have $\kappa^2/\gamma_b \rightarrow \infty$ and hence $\tilde{a} \rightarrow 0$, the hallmark of EIT. However, because our system involves two states that are both coupled to the incident light we would have the additional term $n\tilde{E}_0$:

$$\begin{pmatrix} \tilde{a} \\ \tilde{b} \end{pmatrix} = - \begin{pmatrix} \delta + i\gamma_a & \kappa \\ \kappa & \delta + i\gamma_b \end{pmatrix}^{-1} \begin{pmatrix} g\tilde{E}_0 \\ n\tilde{E}_0 \end{pmatrix}$$

Where n is a geometric parameter indicating how strong the state $|b\rangle$ couples to the incident light. Solving for both \tilde{a} and \tilde{b} with $\delta = 0$ we get $\tilde{a} = (ig - \kappa n/\gamma_b)\tilde{E}_0/(\gamma_a + \kappa^2/\gamma_b)$ and $\tilde{b} = (-\kappa g/\gamma_b + in\gamma_a/\gamma_b)\tilde{E}_0/(\gamma_a + \kappa^2/\gamma_b)$. If the damping rate γ_b is significantly smaller than other parameters we can neglect the terms without γ_b in the denominator and we get $\tilde{a} \rightarrow -(n/\kappa)\tilde{E}_0$ and $\tilde{b} \rightarrow (-\kappa g + in\gamma_a)\tilde{E}_0/\kappa^2$, and because neither goes to zero this system does not show the hallmark property of EIT.

Although our system is similarly described by model of coupled resonators, the objective of our system and that of systems in the context of EIT are different. Our system focuses on the dynamic change of the system (in our case the Fermi level change) and the use of the avoided crossing that occurs between the two resonances to achieve dynamic phase tuning. EIT systems, on the other hand, utilize the phenomenon of EIT itself for spectroscopy and sensing purposes (for example see *Nano Lett.* 8, 3983-3988 (2008)) or for slow light applications derived from the dispersive characteristic of EIT (see *Phys. Rev. Lett.* 101, 047401 (2008), *Nature* 397 594-598, (1999)).

6. Before conclusion, in the section of EF sweeping (line245-260), this reviewer expected to see 3π phase change in the text (shown in Fig.5, but hard to read the graph). To remove the possible confusion, it is suggested to clearly write $\sim 3\pi$ phase evolutions are obtained for various u_s values, for the change of EF=0 to EF(u_s). Maybe between line 251 and 252?

As the Reviewer suggested, we have added the following sentence between **line 251** and **252**:

“...On the other hand, the graphene nanoribbon cases in Fig. 5b utilizing the avoided crossing phenomenon all show $\sim 3\pi$ phase modulation capacities...”

7. In Fig.3, it is recommended to overlay the reflectance and phase curves along the line of evolution (the present color scale is hard to quantify) For example, authors may consider reflectance/phase plot (curve) overlaid just inside (or outside) the x axis (Fig. 3b,e,h/Fig3c,f,i), while little more emphasizing the line of tracing (frequency). This will make it easier for the readers to follow/compare the phase evolution and the reflectance spectra, for the un-patterned / ribbon / TCMT analysis.

We would like to point out that originally in the Supplementary Information, Fig. S3 did show something very similar to what the reviewer suggested, with the reflected light amplitude and phase values (at the operating frequency) plotted with respect to the Fermi level for both the graphene ribbon case and the unpatterned graphene sheet case. We have therefore added the values for the theoretical TCMT case as well to Fig. S3 as shown below, and have made sure to refer to Fig. S3 in the main paper for clarity. As for the current Fig 3 in the main paper, the authors believe that the current format is suitable as a major section of the paper expounds details that require the imagery and the format of Fig. 3b, e, h and c, f, i.

Figure R3. Revised Fig. S3.

8. This reviewer may have missed, but it will be better if authors clarify the ranges of amplitude which could be achieved, in terms of theoretical, and practical viewpoint.

We thank the Reviewer for the opportunity to clarify the achievable amplitude range. We realize that although Fig. 5 was intended to illustrate the matter in question, we did not elaborate enough in the text. Therefore, we have added to and modified the first part of the final paragraph before the Conclusion (lines 245-250) as follows (the added text is underlined):

“To demonstrate the viability of the metasurface for realistic levels of graphene quality currently achievable, as well as illustrate the ranges of amplitude achievable, Fig. 5a and 5b show complex reflection amplitude EF sweeps for different graphene mobilities in the metasurface structure with an unpatterned graphene sheet and graphene nanoribbons, respectively. For both cases, the tendency of the reflectivity circles getting bigger in amplitude is evident, due to the system having less dissipative loss and therefore becoming more over-coupled (13). From a theoretical perspective, given that there are negligible material losses in the system, the amplitude will near unity but there will still be small scattering losses from graphene plasmons reflecting off the Si pillars. Considering that there exist dielectrics and metals that have negligible losses in the infrared regime, the achievable amplitude range would be mainly determined by the loss in graphene. We note that the reflection amplitude can be as high as ~ 0.5 even at a poor carrier mobility of $\mu_s=500 \text{ cm}^2/\text{V}\cdot\text{s}$ and can reach ~ 0.8 at $\mu_s=2,000 \text{ cm}^2/\text{V}\cdot\text{s}$, which is achievable with commercial large-area graphene grown by chemical vapor deposition. With high-quality single crystalline graphene having carrier mobilities exceeding $10,000 \text{ cm}^2/\text{V}\cdot\text{s}$ (35,36,37), the maximum reflection amplitude of the device would reach near unity.”

REVIEWERS' COMMENTS

Reviewer #2 (Remarks to the Author):

The authors faithfully addressed all my questions/comments.

I suggest the publication of this revised manuscript as is.

Response to Reviewers – NCOMMS-21-29703

We thank the Reviewers for their valuable comments. We have carefully revised the manuscript in light of all the received questions and comments that are marked in blue. Our point-by-point responses to them are marked in black.

Reviewer 2

The authors faithfully addressed all my questions/comments. I suggest the publication of this revised manuscript as is.

We thank the Reviewer 2 for the kind consideration of our work.